# Detection and Molecular Phylogenetic-Morphometric Characterization of *Rhizoctonia tuliparum,* Causal Agent of Gray Bulb Rot of Tulips and Bulbous Iris

**DOI:** 10.3390/jof8020163

**Published:** 2022-02-08

**Authors:** Katie Coats, Annie DeBauw, Dilip K. Lakshman, Daniel P. Roberts, Adnan Ismaiel, Gary Chastagner

**Affiliations:** 1Department of Plant Pathology, Washington State University Research & Extension Center, 2606 West Pioneer, Puyallup, WA 98371, USA; Coatskp@puyallup.k12.wa.us (K.C.); annie.debauw@oda.oregon.gov (A.D.); 2Sustainable Agricultural Systems Laboratory, USDA-ARS, 10300 Baltimore Avenue, Beltsville, MD 20705, USA; dan.roberts@usda.gov (D.P.R.); ed.ismaiel@usda.gov (A.I.)

**Keywords:** *Sclerotium tuliparum*, *Ceratobasidium*, quantitative PCR, soil-borne pathogen, sclerotia, ITS2 secondary structure, phylogeny, basidiomycete

## Abstract

Gray bulb rot of tulips and bulbous iris is caused by the soil-borne fungal pathogen, *Rhizoctonia tuliparum* (*Rtul*). Sclerotia present in infected bulbs, as well as overwintering sclerotia in soil and field debris, are the primary sources of infection. A method for accurate and sensitive detection of *Rtul* from soil and infected bulbs, and estimation of inoculum threshold levels, is needed for the management of disease caused by this pathogen. We designed a unique set of primers targeting the ITS2 region of the *Rtul* genome and developed a highly sensitive quantitative PCR (qPCR)-based method for *Rtul* identification using these primers, where the threshold of detection was approximately 1 fg *Rtul* DNA. The assay was more sensitive with sclerotia collected from the field (natural) than with those grown in the lab, and more sensitive with natural-light than natural-dark sclerotia. Also, the detection method was more sensitive when sclerotia were extracted from soil than from bulb tissue. The qPCR method was highly specific, as no PCR amplification was detected when genomic DNA from 62 non-*Rtul Rhizoctonia* isolates from a wide range of anastomosis groups were tested. To understand the evolutionary relationships and genomic diversity of *Rtul*, we performed phylogenetics of the ITS1-5.8S-ITS2 region and ITS2-molecular morphometric characterization (MMC) of *Rtul* isolates. The three *Rtul* isolates whose ITS sequences were available in GenBank formed a distinct phylogenetic clade with *Ceratobasidium anceps* as the nearest relative. Furthermore, MMC analysis revealed genetic divergence among these three *Rtul* isolates.

## 1. Introduction

Gray bulb rot of tulips and bulbous iris is caused by the soil-borne fungal pathogen *Rhizoctonia tuliparum* (*Rtul*) [Whetzel and Arthur; Synonym: *Sclerotium tuliparum* (Kleb)] [1,2,3,4,5,6]. The disease was initially described by Wakker [7] in the Netherlands, and the pathogen was described by Whetzel and Arthur [8]. Gray bulb rot is primarily a problem in field or landscape-grown tulips and irises, but the pathogen has also been reported in daffodil, crocus, gladiolus, and lily [9]. The fungus forms felty masses between the scales, and infected bulbs rot from the top down. Both the bulbs and roots turn grayish in color and progressively dry and shrivel at the site of infection. Infected shoots either fail to emerge or die shortly after infection during late winter. If a plant does grow, it is usually slow to develop and withers and dies before flowering [9]. Diseased plants tend to occur in patches in field and landscape plantings, but this disease can completely destroy the crop after several rotations if unmanaged.

Other than volunteer plants, overwintering sclerotia of *Rtul* residing in soil and host debris are important sources of inoculum. Sclerotia range in size from 1–10 mm in diameter and are formed on infected plant material. They turn from white to dark brown or black as they ripen and can be found on bulbs and in surrounding soil. According to Coley-Smith et al. [10], about 10% of the sclerotia can remain viable in soil for at least ten years. Sclerotia usually remain attached to soil and plant debris when the bulbs are dug, allowing them to be easily spread from field to field by cultivating equipment. The pathogen can also be spread by the movement of infected planting stock.

Currently, there are no reliable and rapid assay methods to assess inoculum levels of *Rtul* in the soil and on infected bulbs [11,12,13]. The ability to quantify inoculum levels of *Rtul* in the soil and on planting stock, coupled with information on inoculum threshold levels necessary for disease development, may allow for the development of predictive diagnostic tests to identify high-risk fields and planting stocks where inoculum of this pathogen is above threshold values [14,15,16,17]. Access to this type of information would assist growers in making management decisions relating to the application of chemical or other control measures and potentially reduce the use of fungicides in the production of tulips and iris [13]. Therefore, we developed a sensitive and specific quantitative PCR (qPCR) assay [18] to detect *Rtul* and tested its effectiveness with bulb and soil samples.

The nuclear status and the teleomorph of the anamorphic fungus *Rtul* have not been unambiguously resolved; isolates from the UK [19] and Netherlands [20] having been reported to be binucleate, potentially placing *Rtul* as *Ceratobasidium* sp. Therefore, we determined the phylogenetic status of *Rtul* within the *Ceratobasidiaceae* using the ribosomal internal transcribed spacer (ITS1-5.8S-ITS2). To gain information on the evolutionary relationships and genomic diversity of *Rtul*, we also characterized the ITS2-primary and secondary structure-based molecular morphometrics (MMC) of isolates currently available in GenBank [21,22].

## 2. Materials and Methods

### 2.1. Fungal isolates

The fungal isolates used in this study are listed in Table 1. Isolate I-399 of *Rtul* was obtained in 1998 from infested soil from a commercial bulbous iris field near Mount Vernon, WA. The other *Rtul* isolates were collected in 2008 and 2012 from symptomatic bulbous irises grown near Puyallup, WA. Agar plugs of *Rtul* were retrieved from long-term storage at 4 °C in water, transferred to potato dextrose agar (PDA), and cultures of *Rtul* maintained on PDA at room temperature prior to DNA extraction. The sixty-two non-*Rtul* isolates used for specificity testing in qPCR assays were stored on filter paper and held at room temperature prior to DNA extraction.

### 2.2. DNA Extraction from Fungal Isolates, Soil, and Plant Tissue

DNA was extracted from homogenized fungal isolates using a Gentra Puregene Tissue Kit (QIAGEN, Valencia, CA, USA), following the manufacturer’s instructions. Sclerotia and/or mycelia were homogenized (FastPrep-24 Homogenizer, MP Biomedicals, Santa Ana, CA, USA) at maximum speed for one min in 2 mL microcentrifuge tubes containing six 3.5-mm-diameter glass beads (BioSpec Products, Bartlesville, OK, USA).

Soil samples were collected from the top 10 cm of a Puyallup fine sandy loam field site at the Washington State University Research and Extension Center in Puyallup, air dried, passed through a 2-mm sieve, and stored at room temperature. Soil was autoclaved at 125 °C for 40 min prior to DNA extraction. DNA was extracted using the Wizard Magnetic DNA Purification System for Food (Cat. No. PRFF3750; Promega, Madison, WI, USA) with Lysis Buffer A replaced by Soil Extraction Buffer (SEB; 120 mM sodium phosphate buffer, pH 8, 2% CTAB, 1.5 M NaCl, 2% antifoam B emulsion). Soil samples were mixed to homogeneity and triplicate 4-g samples weighed and added to 50 mL tubes containing 16 mL SEB and 5 mL 3.5-mm-diameter glass beads. Samples were then homogenized at maximum speed for one min and centrifuged at 2000× *g* for three min. Supernatant (500 µL) was transferred to 2 mL microcentrifuge tubes containing 250 µL Buffer B (Cat. No. Z3191, Promega) using 200 mg of starting material (manufacturer’s instructions were followed starting from Step 3). DNA samples were stored at 4 °C until qPCR analysis.

Tulip bulbs (*Tulipa gesneriana* var. Advent) from a Washington farm were harvested in 2013. Bulbs were surface sterilized with a 0.825% hypochlorite solution, rinsed twice with water, and homogenized to a textured paste consistency. Tulip tissue was kept frozen at –20 °C prior to DNA extraction. DNA was extracted with the Wizard Magnetic DNA Purification System for Food with SEB modification, as described above.

### 2.3. Primer Design and qPCR Assay for R. tuliparum

Sequence of the ribosomal DNA ITS region of the *Rtul* isolates (GenBank accessions KX767078, KX767079, KX767080) were compared among themselves and to a variety of *Rhizoctonia* sequences available in GenBank. Several sequence spans were identified that were conserved within the *Rtul* isolates but variable in other *Rhizoctonia* species. Three sets of primers and TaqMan^®^ MGB probes potentially specific to *Rtul* were designed using Primer Express Software Version 3.0 (Applied Biosystems, Foster City, CA, USA) and tested. The primer/probe set selected for further experimentation was designated as “*Rtul*” and amplified a 100-bp fragment of ITS (Table 2; Appendix A). The *Rtul* hybridization probe was labeled at the 5′ end with tetrachlorofluorescein (TET™) and at the 3′ end with a minor groove binder nonfluorescent quencher (MGB-NFQ) following the manufacturer’s instructions (Applied Biosystems, Bedford, MA, USA). qPCR was carried out at the default cycling parameters of 50 °C for 2 min, 95 °C for 10 min, 40 cycles at 95 °C for 15 s, and 60 °C for 1 min on a 7500 Real Time PCR System (Applied Biosystems) using Sequence Detection Software Version 1.3.1. An exogenous amplification control (EAC) of salmon testes DNA spiked into each reaction to measure potential amplification inhibition was detected by Sketa primers and a Sketa TaqMan^®^ probe [23] labeled at the 5′ end with fluorescein (6-FAM) dye and at the 3′ end with a tetramethylrhodamine (TAMRA™) quencher (Thermo Fisher Scientific, Waltham, MA, USA). The qPCR mixture (25 μL) included 12.5 µL TaqMan^®^ Universal Master Mix II no AmpErase UNG; 100 nM each *Rtul* primers, *Rtul* probe, EAC Sketa primers and EAC Sketa probe; and 0.04 ng µL^−1^ salmon testes DNA (Cat. No. D7656, Sigma Chemical Co., St. Louis, MO, USA). Standard curves were generated based on linear regression of C_t_ values produced by a dilution series of I-399 *Rtul* DNA containing 7.7 × 10^−2^ to 7.7 × 10^−6^ ng DNA. *Rtul* DNA quantity in unknown samples was calculated by applying their cycle threshold (C_t_) value as the independent variable in the resulting linear regression equation. Sterile water in place of the DNA template was run as the negative control. To determine the lowest threshold of detection with the *Rtul* primer/probe set, the concentration of DNA sample extracted from a pure culture of *Rtul* (DGF2-VV) was estimated with a fluorometer (Thermo Fisher Scientific) and used to make a 1:10 serial dilution series from 1.2 ng μL^−1^–0.12 fg μL^−1^ of DNA in the *Rtul* qPCR assay (Figure 1). The amplification efficiency (E) was calculated using the formula: E = 10(−1/slope) – 1 (Ruijter et al. [24]). DNA samples, including standards and controls, were assayed in triplicate (*n* = 3).

### 2.4. Detection of Lab-Grown and Field-Collected (Natural) Sclerotia of R. tuliparum

Lab-grown sclerotia were prepared by transferring small cuttings of I-399 *Rtul* culture, containing mycelia and sclerotia, onto 10-cm-diameter PDA plates and incubating at 23 °C in the dark. Sclerotia were removed from cultures on PDA plates with a sterile pipet tip after 10 weeks and stored at room temperature. Natural sclerotia were obtained by manually sieving air-dried field soil (Puyallup fine sandy loam) which was collected 9 months after iris bulbs were planted in a field site that had been infested with *Rtul* inoculum on rice grains. Sclerotia were then stored in a vial at ambient temperature. Natural sclerotia ranged from brown to black in coloration and were categorized as light-colored or dark-colored (Appendix A). Due to the variability of sclerotia size and weight, 189 lab-grown sclerotia were weighed to determine the average weight of one sclerotium to be 1.78 mg. The amount of sclerotial material used in each experiment was a weight equivalent based on this number. DNA extracted from six replicates of eight (14.25 mg) lab-grown, natural-light, and natural-dark sclerotia were analyzed in triplicate by *Rtul* qPCR undiluted and diluted with sterile water 1:10, 1:100, and 1:1000. Experiments were repeated and combined prior to analysis as there were no significant differences in results.

### 2.5. Detection of R. tuliparum in Soil and Plant Samples

For detection of *Rtul* in soil, the weight equivalent of one, two, four, eight, 16 and 32 lab-grown sclerotia were added to 4 g soil (Puyallup fine sandy loam field soil) samples in triplicate, prior to DNA extraction. DNA samples were diluted 1:1000 and analyzed with the *Rtul*/Sketa qPCR method described above in triplicate and the entire test repeated and results combined prior to analysis.

For the detection of *Rtul* in tulip tissue, the weight equivalent of eight lab-grown sclerotia was added to the weight equivalent of 1/8, 1/4, and 1/2 tulip bulb in triplicate, and DNA was extracted as described above. The tunic was removed from thirty bulbs prior to determining mean bulb weight to be 20.78 g. DNA samples were analyzed by *Rtul*/Sketa qPCR and compared to the equivalent treatment of sclerotia in the absence of bulb tissue.

### 2.6. Phylogenetic Analyses

The *Rtul* ITS sequences of our three GenBank submissions (GenBank accessions KX767078, KX767079, KX767080) were found to be identical through alignment and henceforth will be considered as the same haplotype, accession KX767078 being the largest of the sequences. Accession KX767078 was subjected to a BLAST search using the National Center for Bioinformatics Institute (NCBI) website. Sequences with high homology to KX767078, along with ITS sequences from known species of *Ceratobasidum, Rhizoctonia solani* (*Thanatephorus cucumeris*), and *Waitea circinata* were obtained in FASTA format, aligned using Clustal X [25], and manually adjusted using Mesquite editing software [26]. The final alignment file contained sequence from 34 isolates. Phylogenetic trees were constructed using Bayesian inference (BI) and maximum parsimony (MP) analyses. The Bayesian tree was obtained using MrBayes, Version 3.2.7 [27], using the general time reversible (GTR + I + G, nst = 6) model with gamma distribution and invariable sites with default priors. Four chains and one million generations were run with a tree selected every 1000 generations. The initial 25% of trees were discarded during the burn-in phase. The remaining trees were used to produce a 50% majority rule consensus tree. This tree was edited and posterior probabilities above 0.95 placed above the branches. The MP tree was obtained with PAUP Version b10 software [28] using a heuristic search with a starting tree obtained via 1000 random stepwise addition sequences, and tree-bisection-reconnection as the branch-swapping algorithm, with MULTrees in effect. Bootstrap values were calculated with 1000 replicates. For both the Bayesian and MP trees, *W. circinata* var. *agrostis* was used as an outgroup based on the distance of the isolate from the rest of the isolates in the tree.

### 2.7. Morphometric Analyses of the ITS2 Region

For ITS2 structure analysis, the ITS1-5.8S-ITS2 sequences were retrieved for *C. anceps* isolates CBS 152.32 (GenBank accession MH855251) and CBS 152.32 (AJ427402), and three *Rtul* isolates: isolate I-399 (KX767078), the European isolate 29,792 (EU191041), and isolate from New York CBS206.25 (MH854847) [29]. The corresponding ITS2 sequences were identified utilizing the annotation utility of the ITS2 database [30]. Secondary structure was predicted with the prediction utility of the ITS2 database [30] and manually analyzed for morphometric features like compensatory base change (CBC), hemi-compensatory base change (hCBC), indels of base pairing, pseudoknot formation, and conserved loop sequences. Percent GC was estimated utilizing tools from PAUP Version b10 software [28].

## 3. Results

### 3.1. R. tulipia qPCR Detection Assay Sensitivity and Specificity

Analysis of the serial dilution standard curve showed a linear relationship (R^2^ = 0.994) between the log of the DNA concentration of *Rtul* and the cycle threshold (C_t_) value over the seven most concentrated DNAs (i.e., 7.7 × 10^−2^ to 7.7 × 10^−6^ ng) (Figure 1). The least concentrated DNA sample used in the standard curve, at 0.12 fg μL^−1^, was not detected and most likely indicates that the threshold of detection for this assay was approximately 1 fg *Rtul* DNA. For determination of assay specificity, 62 non-*Rtul* and six *Rtul* isolates of *Rhizoctonia* (Table 1) were analyzed to ensure that amplification occurs for all *Rtul* isolates and does not occur for any non-*Rtul* isolate. Tested isolates were from *R. solani* anastomosis groups (AGs) 2-1, 3, 4, 5, 8, and 9, and *Ceratobasidium* isolates from AGs A, E, and I, as well as *W. circinata* and several unidentified *Rhizoctonia* species. No amplification was observed with the 62 non-*Rtul* DNA samples, and C_t_ values between 13.93 and 33.51 were observed for various *Rtul* DNA samples (Table 1).

### 3.2. Detection of Lab-Grown and Natural Sclerotia

Analysis of dilutions of DNA extracted from three types of sclerotia (lab-grown, natural-dark, and natural-light) revealed that the *Rtul* detection assay was more sensitive, in that it more efficiently detected *Rtul* DNA, when the DNA was diluted in the range of 1:10 to 1:1000 before qPCR (Figure 2). The Sketa internal amplification control confirmed that amplification was suppressed at higher DNA concentration levels. Three replicates each of DNA extracted from the weight equivalent of eight lab-grown, natural-dark, and natural-light sclerotia were diluted 1:1000 and analyzed by the *Rtul* qPCR assay to determine which type of sclerotia yielded more *Rtul* DNA (Figure 3). The natural-light sclerotia yielded an average of two and seven times more DNA than the natural-dark and the lab grown sclerotia, respectively.

### 3.3. Detection of R. tuliparum in Soil

Since sclerotia overwinter in soil, the utility of the qPCR method was evaluated by determining the amount of *Rtul* DNA from various numbers of sclerotia extracted from fixed quantities of soil. Weight equivalents of one, two, four, eight, and 16 lab-grown sclerotia did not yield significantly different *Rtul* qPCR results when extracted alone or in 4 g soil (Figure 4). In contrast, the weight equivalent of 32 sclerotia was detected more efficiently in soil; about 62% more *Rtul* sclerotial DNA was detected in soil compared to pure sclerotia not mixed with anything. The detection of lab-grown sclerotia became unreliable when the number of sclerotia per gram of soil was less than two.

### 3.4. Detection of R. tuliparum in Bulbs

To determine the detection limit of the *Rtul* assay using infected bulb tissue, an increasing amount of tulip bulb tissue was extracted with eight lab-grown *Rtul* sclerotia. An inverse sensitivity of detection of *Rtul* was observed when sclerotial DNA was extracted in the presence of increasing amounts of bulb tissue. At the maximum amount tested, one-half a tulip bulb, detection of DNA from eight lab-grown sclerotia decreased by an average of 84% (calculated as 100 – [100 × 0.45 bulb fraction/2.8 ng recovered DNA]), indicating possible carryover of PCR inhibitors in sample preparations in the presence of bulb tissues (Figure 5).

### 3.5. Phylogenetic Analyses

The Bayesian and MP trees were essentially identical in topology; therefore, we only present the Bayesian tree. The bootstrap support values of 75% or higher from the MP analyses are indicated on the Bayesian tree with the symbol (*) under the branches (Figure 6). The Bayesian tree showed that all three geographically separated isolates of *Rtul* [KX767078 (Washington, DC, USA), EU191041 (Europe) and MH854847 (Ithaca, NY, USA)] formed a monophyletic group with high posterior probability and bootstrap values. The clade had a strong sister relationship with the clade of two *C. anceps* isolates, with posterior probability of 0.99 and bootstrap value of >70%. No further relationship can be inferred from the tree due to lack of phylogenetic support with other clades.

### 3.6. Morphometric Analyses of the ITS2 Region

The ITS2 sequence length was 237 bases for *C. anceps*, and 238 bases for each of the three *Rtul* accessions. The GC content of the ITS2 region was 42.6% for *C. anceps*, and 46.2%, 45.8%, and 45.8% for the isolates of *Rtul* with GenBank Accessions KX767078, EU191041, and MH854847, respectively (Appendix A).

In comparison with the secondary structure of the ITS2 region of *C. anceps* isolate CBS 152.32 (MH855251), the ITS2 secondary structure of *Rtul* isolate I-399 (KX767078) had a CBC and a hCBC in helix II, lost three base pairs in helix III, and lost one base pair in helix IV (Figure 7; Appendix A). The presence of the CBC, hCBC, and a loss/gain of base pairs amongst the two species is commensurate with two distinct biological species [31]. This point is supported by the fact that they formed two distinct phylogenetic clades with strong bootstrap value at the primary ITS2 sequence level (Figure 6). However, phylogenetically, the two species are closer to one another than the many other species of *Ceratobasidium*. The *C. anceps* and the three isolates of *Rtul* have maintained a conserved AAGT in the terminal loop of helix III. The conserved triplets of AAT in helix I, CAC in helix II, and TAG in helix IV were observed in respective ITS2 terminal loops of *C. anceps* and the three *Rtul* isolates. (Figure 7; Appendix A).

In comparison with the ITS2 secondary structure of the *Rtul* Washington isolate I-399 (KX767078), the corresponding structures of European isolate 29,792 (EU191041) only have a point mutation in the loop region (transversion, C → A) of helix III, resulting in the gain of a base pair (A-T) (Figure 8; Appendix A). However, in comparison with the ITS2 secondary structure of isolate I-399, the ITS2 secondary structure of isolate CBS206.25 (MH854847) showed structural changes in helices II, III, and IV. Thus, a G→C transversion in helix II resulted in the loss of a G-C base pairing in helix II. In helix III, (i) a transversion in the loop region (C→A) resulted in the gain of a base pair (A-T), (ii) an A→C transversion resulted in the loss of a base pairing (A-T) needed for a single base pair pseudoknot-like structure, and (iii) a C→G transversion resulted in the net loss of a C-G base pair in helix III. Therefore, a net loss of one base pair was observed in helix III of isolate CBS206.25. A missing pseudoknot in helix III of the CBS206.25 isolate of *Rtul* may be a significant genetic shift. Moreover, a G→A transition in helix IV of the latter isolate resulted in the loss of a C-G base pair.

We also noted that the single base pair pseudoknot-like structure was formed in helix III of *Ceratobasidium* AGs A, Ba, Bb, B(o), C, G, H, I, L, Q, uninucleate *Rhizoctonia* 1(UNR1) isolates (Appendix A), *C. anceps, C. fluccosa, R. alpine, R. butini, C. pseudocornigerum,* and *C. noxium* isolates which phylogenetically belonged in the extended clade containing the *Rtul* isolates (Figure 6). Interestingly, a pseudoknot-like structure did not form in the ITS2 helix III from AG groups E, Fa, Fb, K, P, O, R, S, U, and UNR2 accessions AF200515 and AF200517 (Appendix A).

## 4. Discussion

We developed a robust TaqMan^®^ real-time qPCR assay to specifically detect and quantify *Rtul* DNA based on a unique sequence of the internal transcribed spacer region. Specificity to *Rtul* with the exclusion of other *Rhizoctonia* genera, species, AGs, and subgroups was established by testing the assay on nontarget isolates, of which none amplified. The *Rtul* PCR primers and the probe exactly matched the *Rtul* ITS2 sequence. Moreover, the NCBI BLAST search (dated 14 January 2022) of the *Rtul* probe showed 100% sequence homology with ITS sequence region of *C. pseudocornigerum* (MH861653), *C. angustisporum* (NR_154601), and several species of mycorrhizal *Ceratobasidium*, mostly from orchid hosts (Appendix A). The *Rtul* probe also matched 13 out of 17 bases (data not shown) with its close phylogenic relative, *C. anceps* (Figure 6), which is a pathogen of persimmon (*Diospyros kaki*) and tea [32]. Because of host differences, the presence of above pathogens in proximity to *Rtul* pathogen would be remote. Therefore, the qPCR assay should be considered specific for *Rtul* when tested with soils from iris fields. All *Rtul* samples tested were amplified, and a dilution series of *Rtul* DNA revealed a limit of detection of 1 fg DNA. The *Rtul* qPCR assay presents a signification advance in molecular diagnostics of the pathogen and is appropriate for identification of isolates or sclerotia, as well as detection of the pathogen in soil or plant tissue.

Most *Rhizoctonia* pathogenicity studies use inoculum grown on culture plates, which produce sclerotia which are slightly different in appearance from those naturally occurring in soil. In our observation, lab-grown sclerotia are brown and corky in texture, while natural sclerotia have a darker and harder outer rind. In addition, natural sclerotia range from light brown to black in coloration. Because the intended application of the qPCR assay is to detect natural sclerotia, but validation experiments were performed with lab-grown sclerotia, the assay’s ability to detect *Rtul* DNA in both naturally found and lab-grown sclerotia was compared. The sensitivity of *Rtul* detection from light brown sclerotia was two-fold and seven-fold higher then corresponding natural black sclerotia and lab-grown sclerotia, respectively; thus, confirming the suitability of the qPCR assay to detect *Rtul* in the field.

The presence of soil did not affect the efficiency of DNA extraction or detection of sclerotia using qPCR, as observed by Guo et al. [33] for the detection and quantification of *R. cerealis* in soil. In fact, sclerotial DNA extraction in the presence of soil seemed to remove or neutralize PCR inhibitors. Fungi sequester many secondary metabolites in sclerotia that act as a chemical defense system against competitors, fungivorous predators and parasites [34], although the effect of such chemicals on PCR has not been investigated [35].

The ability to detect *Rtul* DNA decreased with increased tulip bulb tissue, such that the equivalent of almost ten sclerotia would need to be present if one whole tulip bulb was extracted. This limited sensitivity has the potential to limit the effectiveness of the *Rtul* assay in quantifying inoculum levels in planting stock. It is likely the decreased sensitivity of the *Rtul* assay associated with increased bulb tissue may be due to the presence of qPCR inhibitors in the dormant tulip bulb tissues [36]. A variety of protocols have been developed to remove inhibitors from nucleic acids before PCR [37]. Additional research is needed to determine if removal of potential inhibitors from tulip bulb tissues to increase the sensitivity of the whole bulb assay, and thereby maximize the likelihood of detecting bulb infections, would be beneficial. Moreover, Tellenbach et al. [38] reported increased sensitivity of qPCR detection from sclerotial DNA when probes were made from a single-copy gene as opposed to a multicopy gene like ITS. This enhancement effect was attributed to dilution of PCR inhibitors after the first amplification step. With the genome information of *Rtul* potentially available, future enhancement of pathogen detection from sclerotial DNA may be possible with probes designed from single copy genes [39].

We also conducted ITS (ITS1-5.8S-ITS2)-based phylogenetic characterization and ITS2-based molecular morphometric characterization of three geographically distinct *Rtul* isolates and demonstrated divergent relationships among them as well as their taxonomic position with fungi belonging to *Ceratobasidiaceae*. The Bayesian inference of phylogeny revealed that the three isolates of *Rtul* investigated have originated as a distinct clade, but closer to another distinct clade consisting of a binucleate *Rhizoctonia*, *C. anceps*. Besides, the *Rtul* isolates were more distant from other *Ceratobasidium* spp., including *Ceratobasidium* AGs A to U. This hypothesis is well-supported from the ITS2 molecular morphometric analyses. ITS2 of *C. anceps* has lower GC content and is one nucleotide shorter than those of the *Rtul* isolates (Appendix A). Also, secondary structures of the two species have at least one CBC and one hCBC in helix II, indel of three base pairs in helix III, and indel of one base pair in helix IV (Figure 7).

Recent studies have demonstrated that ITS2 sequence evolves very quickly amongst eukaryotes, making it an excellent marker that is widely used in low level phylogenetic analyses and DNA barcoding of organisms, including fungi [40,41,42,43]. By the same token, any hypervariable feature may also be problematic for investigating deeper phylogeny. This could be overcome by its conserved secondary structure. The conservation of ITS2 secondary structure is a critical component of rRNA processing [41]. Thus, any compensatory base change (CBC) and hemi-compensatory base change (hCBC) in the two most conserved helices, namely helix II and helix III, have been correlated with delineation of species [44]. Speciation in eukaryotes, particularly amongst plants and fungi, has been correlated with even one CBC in the relatively conserved pairing positions of the ITS2 transcript secondary structure [45]. The probability to discriminate between two species with a single CBC between the members of an ITS2 sequence-structure pair has a confidence of 93% [31,45]. On the other hand, if there is no CBC, then individuals may belong to the same species with 76% confidence [45]; Wolf et al. [31]. Molecular morphometric analyses utilize the ITS2 primary sequence as well as secondary structural constraints, homologous locations, indels and substitution to distinguish closely related fungal and other species [21,22]. As a proof of concept for *Rhizoctonia* species, Ahvenniemi et al. [46] showed that the potato (PT) and tobacco (TB) subgroups of *R. solani* AG 3 differ in the ITS2 secondary structure helix I by two CBCs. Moreover, the CBCs in potato strains of AG 3 distinguish them from other AGs of *R. solani*, confirming the hypothesis that the latter is functionally a composite species. Interestingly, our investigations on the ITS2 molecular morphometrics of the three geographically distant *Rtul* isolates demonstrated that major genetic shifts have taken place resulting in significant alteration of respective ITS2 secondary structures. Thus, although the ITS (ITS-5.8S-ITS2) primary sequences of all the three isolates of *Rtul* formed a well-supported clade in the phylogenetic tree (Figure 6), molecular morphometric analyses clearly differentiated them as three potentially distinct sexual species within *R. tuliparum*, sensu lato, per Müller et al. [45] and Wolf et al. [31] (Figure 8; Appendix A), and call for further investigation. Moreover, from disease epidemiology and management viewpoints, our findings call for accurate identification of pathogen genotypes as well as determination of sensitivities of genetically diverse isolates of *Rhizoctonia* pathogens to commonly used fungicides [13,43,47,48,49,50].

The significance of the formation of single base pair pseudoknot-like structure in the stem III of ITS2 in *Rtul* and phylogenetically related species (Figure 6; Appendix A) is not understood at present but may serve as an additional criterion to differentiate groups or strains of *Ceratobasidium* spp. Sharon et al. [51] phylogenetically clustered the *Ceratobasidium* isolates AG-A to AG-U in six groups. In this regard, it is interesting to note that the structure was present in group 4 (AG-D), group 5 [AG-B (including its three subgroups) and AG-Q], and group 6 (AG-C, AG-H, and AG-I) and absent in group 3 (AG-E, AG-F, AG-P AG-U, AG-R, and AG-S). Mixed results were also noted because in group 1 (e.g., AG-A has the pseudoknot, but not in AG-K), and in group 2 (e.g., AG-G and AG-L has the pseudoknot but absent in AG-O).

Taken together, both the development of a sensitive qPCR detection method and determination of the genetic diversity of the *Rtul* pathogens as documented in this investigation are significant for disease management. While cultural practices such as deep plowing to bury sclerotia, shallow planting, and flooding help control this disease, treatments such as soil fumigation, fungicide bulb dips or in-furrow or broadcast applications of fungicides to the soil at the time of planting are also often used as methods of disease control [3,4,9,12,52,53]. In this context, additional research is needed to determine soil-borne inoculum thresholds through qPCR, as developed in this investigation, for disease development. Given the spatial distribution of the disease, additional research is also needed to define a sampling protocol that could be used along with this qPCR diagnostic assay to accurately predict the risk of disease development. Furthermore, the presence of genetic diversity of *Rtul* pathogens as determined in this report also calls for assaying differential sensitivity of *Rtul* isolates to various fungicides and finding the most efficient fungicide for each *Rtul* genotype. A sensitive detection method, along with the use of the most effective fungicide, would ultimately give rise to environmentally and economically sound management of gray bulb rot in tulips and bulbus irises [54,55,56].

## Figures and Tables

**Figure 1 jof-08-00163-f001:**
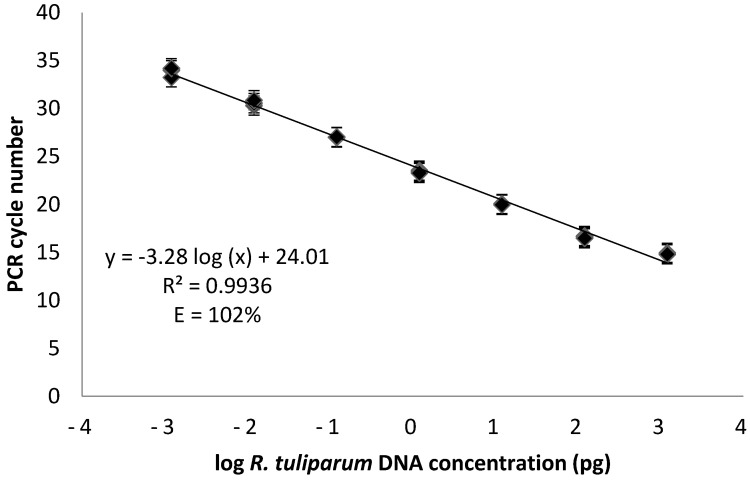
Determining the lowest threshold of detection with the *Rtul* primer/probe set used in this study. 1:10 serial dilution series (1.2 ng μL^−1^–0.12 fg μL^−1^ of DNA) of *Rhizoctonia tuliparum* (DGF2-VV) DNA analyzed in triplicate with qPCR. The log DNA concentration versus cycle threshold was plotted and a regression line drawn for the data.

**Figure 2 jof-08-00163-f002:**
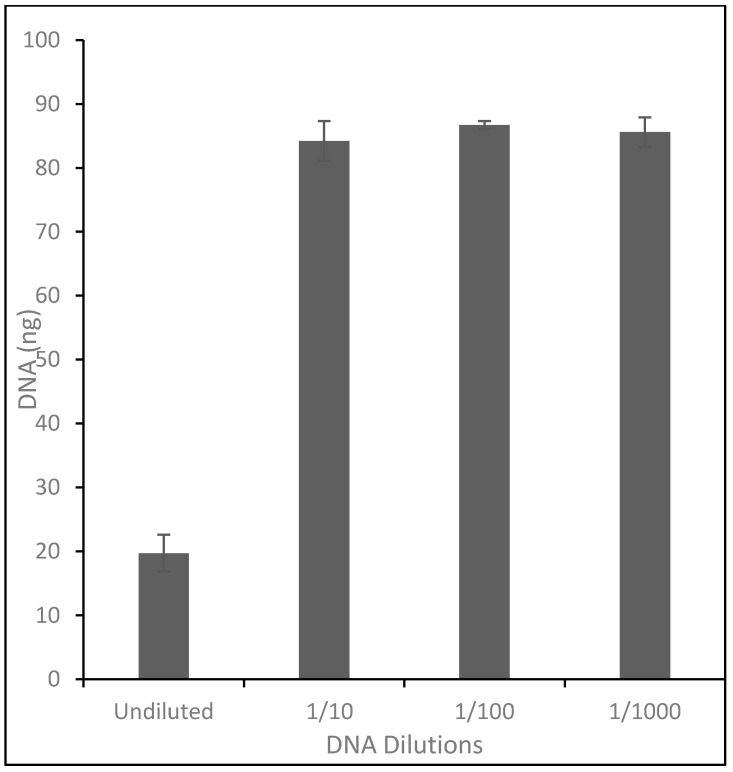
Effect of dilution on the sensitivity of detection of DNA from *Rhizoctonia tuliparum* sclerotia. DNAs extracted from lab-grown sclerotia in triplicate were diluted and analyzed by the *Rtul* qPCR assay, and the detected DNA for each dilution plotted. Error bars represent one standard deviation from the mean. The assays were more precise when the DNA was diluted in the range of 1:10 to 1:1000 before qPCR.

**Figure 3 jof-08-00163-f003:**
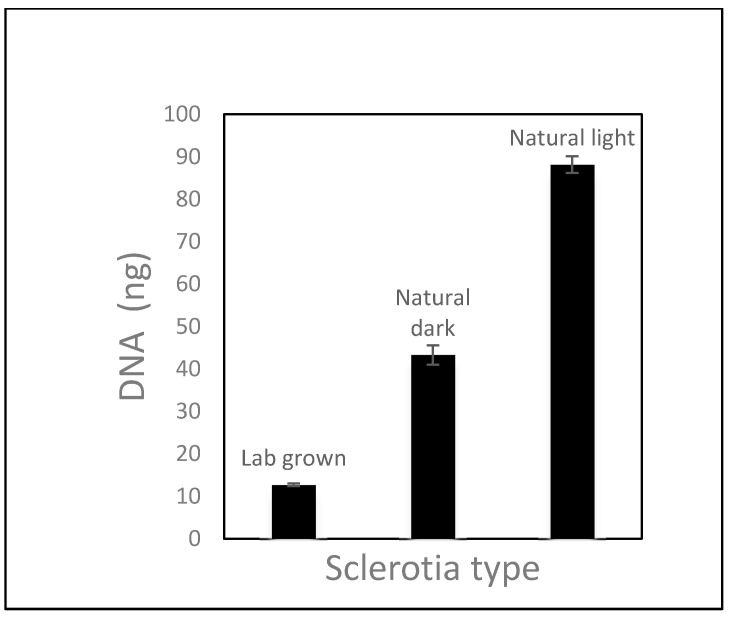
Sensitivity of detection of *Rhizoctonia tuliparum* DNA from lab-grown, natural-light, and natural-dark sclerotia. DNAs were extracted from 14.25 mg sclerotia (equivalent of eight average size sclerotia) of three different types in triplicate diluted 1/1000 and analyzed by qPCR. The detected DNAs for each type were plotted in a bar chart with error bars representing one standard deviation from the mean.

**Figure 4 jof-08-00163-f004:**
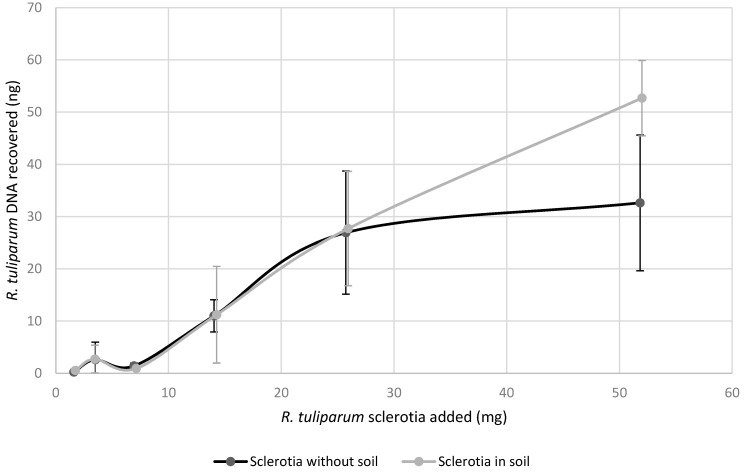
The Rtul qPCR assay was used to evaluate the effect of soil on the detection of Rhizoctonia tuliparum sclerotial DNA. The average amount of R. tuliparum DNA detected when the weight-equivalent of one to 32 sclerotia was added to four grams of soil was fairly consistent, with the exception of the highest amount of sclerotia that yielded more DNA in the presence of soil.

**Figure 5 jof-08-00163-f005:**
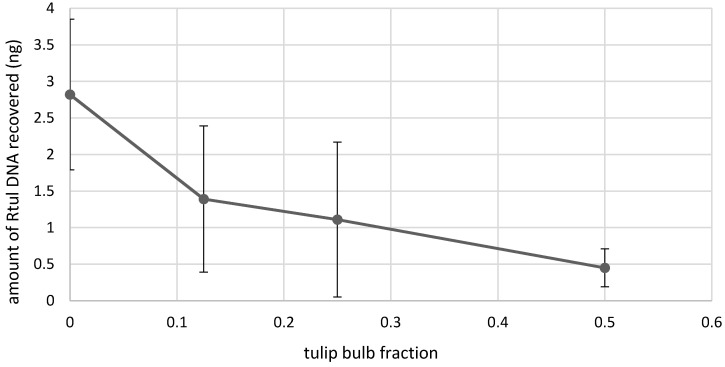
Analysis of the effect of tulip bulb tissue on sensitivity of qPCR detection of *Rhizoctonia tuliparum* DNA. 14.25 mg sclerotia (equivalent of 8 average size lab-grown sclerotia) were added to various fractions of tulip bulb tissue to determine the limit of detection of sclerotial DNA in infected bulbs by *Rtul* qPCR. The amount of *R. tuliparum* DNA detected was compared to the amount of DNA in sclerotia in the absence of bulb tissue. The presence of bulb tissue reduced the ability to detect *R. tuliparum*.

**Figure 6 jof-08-00163-f006:**
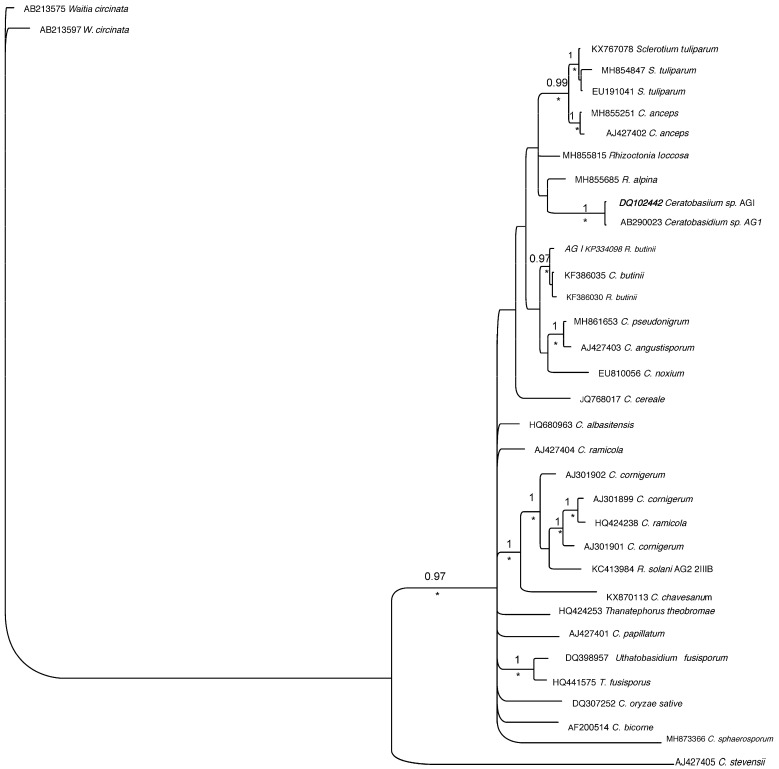
Bayesian tree obtained based on internal transcribed spacer sequences illustrating phylogeny of *Rhizoctonia tuliparum* within closely related species. One million generations with 25% of the initial trees discarded in the burn-in phase were used for analysis. Posterior probabilities >0.95 are shown above the branches. The tree was rooted with two isolates of *Waitea circinata*. The symbol (*) under branches indicates bootstrap support ≥ 70% obtained from parsimony analysis.

**Figure 7 jof-08-00163-f007:**
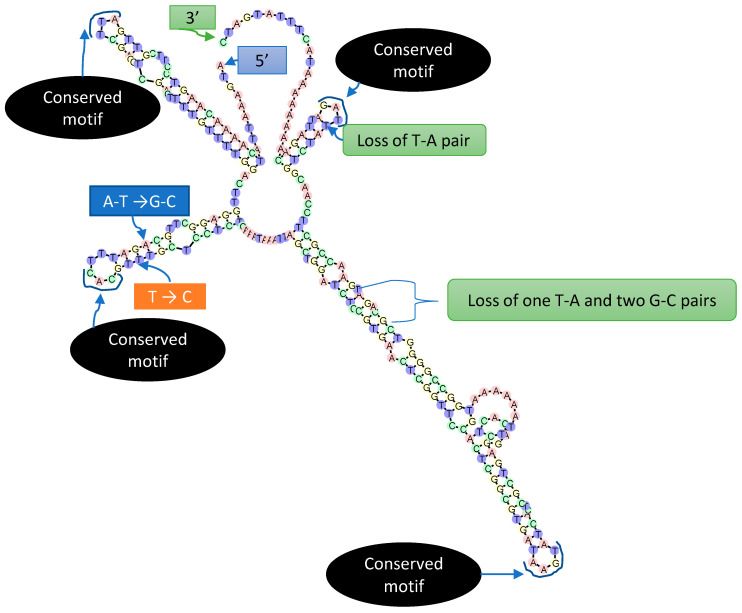
The ITS2 secondary structure of *Ceratobasidium anceps*, isolate CBS 152.32 (GenBank Accession MH855251). as predicted by the ITS2 DB ‘Predict’ utility. The nucleotides are highlighted in green (C), orange (G), violet (T), and pink (A). The structural changes in ITS2 of *Rhizoctonia tuliparum* isolate I-399 (GenBank Accession KX767078) are indicated by compensatory base change (CBC) in the blue box, the hemi-compensatory base change (hCBC) in the orange box, and the loss of base pairs in light green boxes. The conserved terminal loop sequences are marked with curved lines and indicated in black oval boxes in all four helices.

**Figure 8 jof-08-00163-f008:**
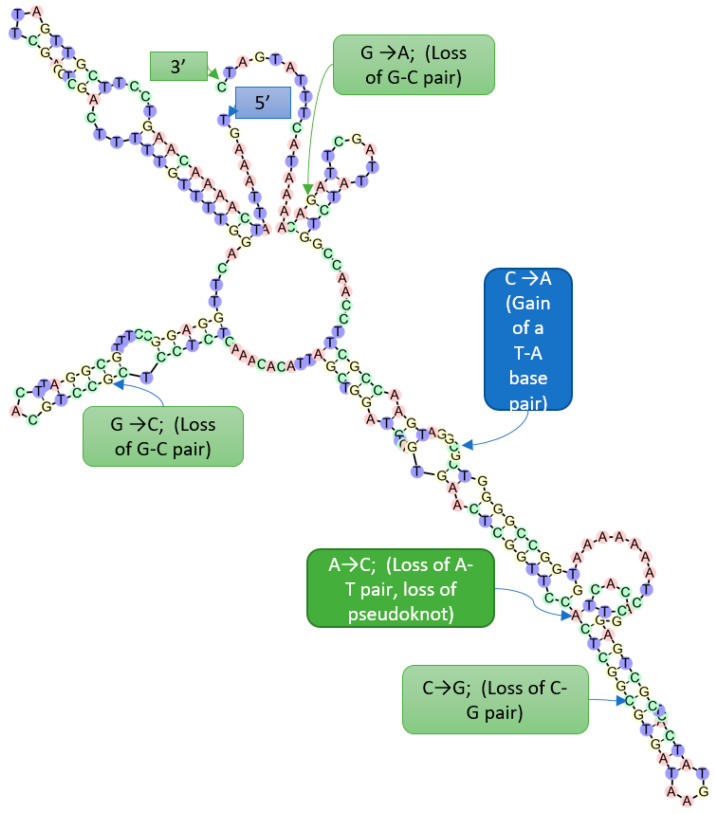
The ITS2 secondary structure of *Rhizoctonia tuliparum* isolate I-399 from Washington State (GenBank Accession KX767078), as predicted by the ITS2 DB ‘Predict’ utility. The nucleotides are highlighted in green (C), orange (G), violet (T), and pink (A). The structural changes in ITS2 of the *R. tuliparum* isolate 29,792 from Europe [EU19104], and the isolate CBS206.25 from New York [MH854847] are indicated as gain of a base pair due to a hemi-compensatory base change (hCBC) in the blue box and loss of base pairs due to hCBCs in the light green boxes. A hCBC resulting in loss of a pseudoknot in the isolate CBS206.25 [MH854847] is indicated in the dark green box.

**Table 1 jof-08-00163-t001:** Fungal isolates used to test species specificity of the TaqMan qPCR assay for *Rhizoctonia tuliparum*, and their resulting *Rtul* cycle threshold (C_t_) value.

Isolate	Species	Host/Source	Origin	C_t_ Value
PC76	*Rhizoctonia solani* AG-2-1	Tulip	Washington	N/A ^a^
Rh060811 ^c^	*R. solani* AG 2-1	Pea	Washington	N/A
Rh070913 ^c^	*R. solani* AG 2-1	Onion	Washington	N/A
Rh070937 ^c^	*R. solani* AG 2-1	Pea	Washington	N/A
Rh070933 ^c^	*R. solani* AG 3	Potato	Washington	N/A
Rh070934 ^c^	*R. solani* AG 3	Potato	Washington	N/A
Rh070935 ^c^	*R. solani* AG 3	Onion	Washington	N/A
Rh070942 ^c^	*R. solani* AG 3	Pea	Washington	N/A
Rh070912 ^c^	*R. solani* AG 3	Onion	Washington	N/A
Rh010901 ^c^	*R. solani* AG 4	Pea	Washington	N/A
Rh070908 ^c^	*R. solani* AG 4	Pea	Washington	N/A
Rh070909 ^c^	*R. solani* AG 4	Pea	Washington	N/A
Rh070910 ^c^	*R. solani* AG 4	Pea	Washington	N/A
Rh070915 ^c^	*R. solani* AG 4	Onion	Washington	N/A
Rh070929 ^c^	*R. solani* AG 4	Onion	Washington	N/A
Rh070939 ^c^	*R. solani* AG 4	Onion	Washington	N/A
Rh070940 ^c^	*R. solani* AG 4	Potato	Washington	N/A
Rh070930 ^c^	*R. solani* AG 5	Potato	Washington	N/A
Rh070931 ^c^	*R. solani* AG 5	Pea	Washington	N/A
Rh070932 ^c^	*R. solani* AG 5	Potato	Washington	N/A
Rh080714 ^c^	*R. solani* AG 8	Wheat	Washington	N/A
Rh010911 ^c^	*R. solani* AG 8	Onion	Washington	N/A
Rh070922 ^c^	*R. solani* AG 8	Soil ^d^	Washington	N/A
Rh070927 ^c^	*R. solani* AG 8	Onion	Washington	N/A
Rh070943 ^c^	*R. solani* AG 8	Pea	Washington	N/A
Rh100557 ^c^	*R. solani* AG 8	Wheat	Washington	N/A
Rh070921 ^c^	*R. solani* AG 9	Pea	Washington	N/A
Rh070938 ^c^	*R. solani* AG 9	Pea	Washington	N/A
Rh090801 ^c^	*Ceratobasidium* AG-A	Onion	Washington	N/A
Rh010913 ^c^	*Ceratobasidium* AG-A	Onion	Washington	N/A
Rh070923 ^c^	*Ceratobasidium* AG-E	Soil ^d^	Washington	N/A
Rh110712 ^c^	*Ceratobasidium* AG-I	Wheat	Washington	N/A
Rh070716 ^c^	*Ceratobasidium* AG-I	Wheat	Washington	N/A
Rh070914 ^c^	*Ceratobasidium* AG-I	Pea	Washington	N/A
Rh010905 ^c^	*Ceratobasidium* sp.	Onion	Washington	N/A
Rh010909 ^c^	*Waitea circinata* (*R. oryzae* grp. 1)	Onion	Washington	N/A
Rh070924 ^c^	*W. circinata* (var. *circinata*)	Soil ^d^	Washington	N/A
Rh070925 ^c^	*W. circinata* (var. *circinata*)	Soil ^d^	Washington	N/A
Rh070936 ^c^	*W. circinata* (var. *circinata*)	Onion	Washington	N/A
Rh070911 ^c^	*Ceratobasidium* sp.	Pea	Washington	N/A
Rh070926 ^c^	*Rhizoctonia* spp.	Soil ^d^	Washington	N/A
Rh070928 ^c^	*Rhizoctonia* spp.	Onion	Washington	N/A
VSP 05-01 A ^c^	*R. solani* AG 4, HG II	Onion	Columbia Basin	N/A
VSP 05-01 B ^c^	*R. solani* AG 4, HG II	Onion	Columbia Basin	N/A
VSP 05-37 A ^c^		Swiss Chard	Washington	N/A
VSP 05-37 B ^c^		Swiss Chard	Washington	N/A
VSP 06-26 B ^c^	*R. solani* AG 4, HG II	Onion	Columbia Basin	N/A
VSP 08-13 A ^c^		Onion	Columbia Basin	N/A
VSP 08-14 B ^c^		Onion	Columbia Basin	N/A
VSP 08-19 A ^c^		Onion	Columbia Basin	N/A
VSP 08-19 G ^c^		Onion	Columbia Basin	N/A
VSP 08-33 B, plt 4 ^c^		Skullcap	Columbia Basin	N/A
06SPFWLA ^c^	*R. solani* AG 3	Spinach	Washington	N/A
VSP 08-33 A, plt 6 ^c^	*R. solani*	Skullcap	Columbia Basin	N/A
VSP 10-20 Tamara 1 ^c^	*R. solani*	Onion	Columbia Basin	N/A
VSP 10-20 Tamara 2 ^c^	*R. solani*	Onion	Columbia Basin	N/A
VSP 10-20 Field 2-2 ^c^	*R. solani*	Onion	Columbia Basin	N/A
VSP 10-20 Field 2-3 ^c^	*R. solani*	Onion	Columbia Basin	N/A
LP 3 ^c^	*R. solani*	Pea	Oregon	N/A
LP 10 ^c^	*R. solani*	Pea	Oregon	N/A
LP 12 ^c^	*R. solani*	Pea	Oregon	N/A
LP 13 ^c^	*R. solani*	Pea	Oregon	N/A
MBL12140	*R. tuliparum*	Iris	Washington	19.60
MBL12141	*R. tuliparum*	Iris	Washington	16.10
MBL12145	*R. tuliparum*	Iris	Washington	27.25
MBL12146	*R. tuliparum*	Iris	Washington	17.84
MBL11082 (DGF2_VV)	*R. tuliparum*	Iris	Washington	33.51
Rh_Tulip	*R. tuliparum*	Tulip	Washington	13.93
I-399 ^b^	*R. tuliparum*	Iris	Washington	
I-399	*R. tuliparum* 1:10	Iris	Washington	18.79
I-399	*R. tuliparum* 1:100	Iris	Washington	21.32
I-399	*R. tuliparum* 1:1000	Iris	Washington	24.90
No template control ^e^	*---*	---	---	N/A

^a^ N/A = no amplification; ^b^ I-399 DNA used for standard curve; ^c^ Cultures provided by Lindsey du Toit, Washington State University, Mount Vernon Northwestern Washington Research and Extension Center; ^d^ Isolated from wheat cultivated soil; ^e^ No DNA template control.

**Table 2 jof-08-00163-t002:** Primers and probes used for the TaqMan^®^ quantitative PCR diagnostic assay for detection and quantification of *Rhizoctonia tuliparum* and the associated exogenous amplification control ^a^.

Primer/Probe Name	Sequence (5′-3′)	Target Region	Study
*R. tuliparum* primers			
*Rtul*_for	CGAGGTCGACTTTTTGTTTTGG	ITS^a^	This study
*Rtul*_rev	CCGAGTGGAACCGAGTTCAC	ITS	This study
*R. tuliparum* probe *R. tuliparum* probe			
*Rtul*_probe	[TET] TTTGCGGATTCACGTCC [MGB-NFQ]	ITS	This study
	Sketa Exogenous Amplification Control (EAC) primers		
SketaF2	GGTTTCCGCAGCTGGG	ITS	[23]
SketaR3	CCGAGCCGTCCTGGTCTA	ITS	[23]
Sketa probe			
SketaP2	[6-FAM] AGTCGCAGGCGGCCACCGT [TAMRA]	ITS	[23]

^a^ ITS, Internal Transcribed Spacer region of ribosomal DNA; TET, tetrachlorofluorescein; MGB-NFQ, minor groove binder nonfluorescent quencher; 6-FAM, fluorescein (6-FAM) dye; TAMRA, tetramethylrhodamine quencher.

## Data Availability

All relevant data generated or analyzed during this study are included in this article.

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
