# Peer review of "Detection and Molecular Phylogenetic-Morphometric Characterization of Rhizoctonia tuliparum, Causal Agent of Gray Bulb Rot of Tulips and Bulbous Iris"

_jof, 2022, doi:10.3390/jof8020163_

Round 1

Reviewer 1 Report

This work provided a nuique set of primers for qRT-PCR detection in R. tuli. It was used to test different kinds of materials, i.e. fungal samples, soil, and plant tissue. This study also performed phylogenetic analyses of different R. tuli and revelaved the ITS2 secondary structure of R. tuli and C. anceps. Here are some questions:

1, In Table 1, more than 60 Rhizocotnia strains were listed. But I can not find the results of specificity testing using these strains. Is this result missed? At L22 and L227, How is the results of primer specificity?  In Table1,  I can not find the Ct value.  Please have a check.

Table1 is more than one page, need a "continued" at the upper left corner.

2, How is the different of three kinds of sclerotia? Can you provide a picture? This work mainly focused on sclerotia, and the results showed difference in different kinds of sclerotia. I suggest you can provide a ptictuce to the redears.

3,  Data in this work need to perform Statistical Analysis.

4,  As in Fig2, I think the results of three kinds of sclerotia need to be shown in Fig.2.  Only lab-sclerotia was presented.

5,  Fig1, Fig5, Only a standard curve was provide. I think it it not enough. How is the results of the targets? Like L268, how to obtained 84% form Fig.5. 

6, The describtion of Fig4 is diffcult to understand when refer to Fig4. Also, without Statistical Analysis, it is hard to determine whether there is significant difference between these two samples.

7, The quality of figures in this work need to be improved.

Author Response

This work provided a unique set of primers for qRT-PCR detection in R. tuli. It was used to test different kinds of materials, i.e. fungal samples, soil, and plant tissue. This study also performed phylogenetic analyses of different R. tuli and revealed the ITS2 secondary structure of R. tuli and C. anceps. Here are some questions:

Response: The authors thank you for those comments and suggestions to improve the manuscript.

1, In Table 1, more than 60 Rhizocotnia strains were listed. But I cannot find the results of specificity testing using these strains. Is this result missed? At L22 and L227, How is the results of primer specificity?  In Table1, I cannot find the Ct value.  Please have a check.

Response: The authors thank the reviewer for asking clarification. We like to mention that the Ct values of R. tuliparum isolates are stated in the Table 1. However, no amplification was noticed within the 35 PCR cycles with other Rhizoctonia isolates, as we have explained in the footnote of footnote of N/Aa.

Table1 is more than one page, need a "continued" at the upper left corner.

Response: The authors thank the Reviewer for pointing out. We could do it, but the settings may change in the proof version. So, it will be better if MDPI Printing Office sets it.

2, How is the different of three kinds of sclerotia? Can you provide a picture? This work mainly focused on sclerotia, and the results showed difference in different kinds of sclerotia. I suggest you can provide a picture to the readers.

Response: A supplemental figure of lab-grown and natural sclerotia is provided.

  1. Data in this work need to perform Statistical Analysis.

Response: We’ve revised the figures to include standard deviation error bars.

  1. As in Fig2, I think the results of three kinds of sclerotia need to be shown in Fig.2.  Only lab-sclerotia was presented.

Response: We have done the effect of dilutions of DNA samples for Q-PCR sensitivity. Subsequently, we have analyzed the sensitivities of the three types of sclerotia for Q-PCR detection in Figure 3. We have shown that the natural-light sclerotia yielded an average of two and seven times more DNA than the natural-dark and the lab grown sclerotia, respectively.

  1. Fig1, Fig5, only a standard curve was provided. I think it is not enough. How is the results of the targets? Like L268, how obtained 84% form Fig.5. 

Response: Figure 5 is now edited to show amount of DNA recovered related to the amount of tulip bulb present. Previously it was the fraction of the total amount of DNA recovered without bulb present compared to with the bulb present and that seemed a confusing way to present the data, although the graphs are proportional to each other. The origin of this number is 0.45/2.82, the average amount of DNA recovered with half a tulip present divided by the amount when no tulip is present. We have inserted this calculation in parenthesis in the sentence stating the 84% decrease if DNA detection.

6, The description of Fig4 is difficult to understand when refer to Fig4. Also, without Statistical Analysis, it is hard to determine whether there is significant difference between these two samples.

Response: Figure 4 is now edited for clarity. Error bars are included in the figure.

7, The quality of figures in this work need to be improved.

Response: We have gone through all the Figures and replaced them with clearer Figures.

Reviewer 2 Report

The manuscript entitled "Detection and Molecular Phylogenetic-Morphometric Characterization of Rhizoctonia tuliparum, Causal Agent of Gray Bulb Rot of Tulips and Bulbous Iris" is well-edited. In addition, its quality is high. The test methods are sufficiently detailed to allow repeatability of the measurements. Also, when using multiple products (e.g. DNA extraction), the manufacturer was followed by their instructions. In general, the test methods are very detailed, but in some cases, they are too detailed. Here is an example: Line 160: Sclerotia were then stored in a vial at ambient temperature on a lab shelf.  Efforts should be made to maintain a balance and provide only really useful information. The novelty of the results is significant. These are the limits of detection of sclerotia from the soil, the phylogenetic results and the newly developed qPCR test for the detection of R.tul DNA. I have some comments that can be found in the version that I uploaded. If I were to look for the weakness of the manuscript, I would say that there are few references made in the last five years. Those older than ten years should be refreshed (with the exception of method descriptions). I fully support the acceptance of the manuscript after a brief minor correction. 
Congratulations to the authors.

Author Response

Congratulations to the authors.

Response: The authors appreciate the Reviewer’s encouragements and appreciation of our research!

Q: There are few references made in the last five years

Response: We could not include more recent research papers because there is a dearth of recent papers relating to the biology, pathology, and management of R. tuliparum. While the reasons are not entirely explicable, we suspect that there is a reduction in the number of pathologists working on diseases of ornamental bulb crops. Other than some recent reviews we already cited (e.g., Lakshman et al., 2019; McGovern and Elmer, 2018), we have added a few more recent review article (Conijn, 2017, Chastagner and Garfinkel, 2020) relating to the subject in the last paragraph in the Discussion.

Chastagner, G., Hanks, G., Daughtrey, M., Yedidia, I., Miller, T., Pappu, H. 2012. Sustainable Production and Integrated Management: Environmental Issues. In: Kamenetsky, R., H. Okubo Eds., Ornamental Geophytes: From Basic Science to Sustainable Horticultural Production. CRC Press, Boca Raton, USA, pp. 363-420. https://doi.org/10.1201/b12881.

Conijn, C. Tulip Diseases. Roodbont Publishers B. V., Zutphen, The Netherlands, 2017; pp. 149.

Chastagner, G.A., Garfinkel, A. Diseases Affecting Ornamental Geophytes and Their Control. In Achieving Sustainable Cultivation of Ornamental Plants, Reid, M. Ed.; Burleigh Dodds Science Publishing, United Kingdom, 2020; pp. 367-414. https://doi.org/10.19103/AS.2020.0066.15.

Q: “FASTA format, aligned using Clustal X [25], and manually adjusted using Mesquite editing software [26].”

Response: Thank you for pointing out. We understand the reviewers logic that those information may not be necessary. However, we humbly state that the procedures cited may be useful for beginners in phylogenetic analyses. So, if there is no strong objection, we would like to keep the sentence.

Q: “Percent GC was estimated utilizing online tools from https://molbiol-tools.ca/DNA_composition.htm.”

Response: Thank you for pointing out. We have cited another source of GC analysis this time (PAUP version b10 software).

Q: “No further relationship can be inferred from the tree due to lack of phylogenetic support with other clades.”

Response: we have changed to another version of the Figure 6 that has the black color instead of gray colored fonts.

Other suggestions directly mentioned in the pdf:

Q: Other minor modifications pointed out in the pdf:

Response: We have also done minor modifications in the text following your suggestions. Thank you!